# Characterization of Elastic Polymer-Based Smart Insole and a Simple Foot Plantar Pressure Visualization Method Using 16 Electrodes

**DOI:** 10.3390/s19010044

**Published:** 2018-12-22

**Authors:** Wangjoo Lee, Seung-Hyeon Hong, Hyun-Woo Oh

**Affiliations:** 1Hyper-connected Basic Technology Research Division, Electronics and Telecommunications Research Institute, Gajung-ro 218, Yusung-gu Daejeon 34129, Korea; wjlee@etri.re.kr (W.L.); hsh6808@naver.com (S.-H.H.); 2Department of computer science, KoreaTech University, Choongjeol-ro 1600, Byungcheong-myun, Cheonan 31253, Korea

**Keywords:** smart insole, CNT, PDMS, plantar pressure visualization, pseudo interpolation scheme

## Abstract

In this paper, we propose a smart insole for inexpensive plantar pressure sensing and a simple visualizing scheme. The insole is composed of two elastomeric layers and two electrode layers where the common top electrode is submerged in the insole. The upper elastomeric layer is non-conductive poly-dimethyl-siloxane (PDMS) and supports plantar pressure buffering and the lower layer is carbon nano-tube (CNT)-dispersed PDMS for pressure sensing through piezo-resistivity. Under the lower sensing layer are 16 bottom electrodes for pressure distribution sensing without cell-to-cell interference. Since no soldering or sewing is needed the smart insole manufacturing processes is simple and cost-effective. The pressure sensitivity and time response of the material was measured and based on the 16 sensing data of the smart insole, we virtually extended the frame size for continuous and smoothed pressure distribution image with the help of a simple pseudo interpolation scheme.

## 1. Introduction

Recently wearable device technologies have become widely researched and some of them have already appeared in the consumer market. The prime aim of the wearable device is gathering the wearer’s condition and movement in a timely way and continuously with newly developed compact devices. A big application of wearable devices is healthcare and sports-related areas. Among these, since the human gait pattern may reveal potential sickness and intrinsic healthcare-related habits, expensive and burdened gait pattern measurement and analyses have been performed in the professional Gait Laboratory. A smart insole is supposed to replace a large portion of the current huge medical equipment [1,2,3]. There are two tracks for composing a planter pressure measuring insole. One is integrating individual pressure sensor(s) such as FSR (force sensing resistor) with insole shape pressure buffering material [4,5,6] and the other is electrically dividing an elastic and pressure sensitive insole into multiple cells using bias and read lines [7]. 

Figure 1 shows a representative case of a conventional foot pressure sensor. FlexinFit is the evolution in the biomechanical and postural analysis, to date tied to the most exclusive use of baropodometric platforms. With more than 400 sensors, the system takes up to four hours. Samsung’s smart insole is a wearable Internet of Things (IoT) device that measures various activities such as activity amount and walking method, transmits data with a smartphone etc. with various sensors attached. A smart insole by using various motion sensors can be used for health care such as walking monitoring and proofreading, moving distance and calorie calculation. The existing technology requires another pressure sensor in addition to shoe products. FlexinFit will be installed inside the shoe as another sensor. Since it is produced by the printing method using the FSR sensors, it has the disadvantage of slipping on the foot when walking. The smart insole by Samsung is made to wear on shoes as a separate product, and when the user puts a shoe on, the user feels uncomfortable. The viewpoint of our research is to keep the function possessed by the existing insole, while using the insole as a sensor. Ultimately, we use facilities and processes to manufacture the insole, but add extra facilities and processes to make the insole a sensor.

Since the latter method needs no soldering or sewing of independent sensors, the fabrication process is simple and cost advantageous. However due to the leakage currents between the physically connected cells, an appropriate cell-to-cell interference removing scheme is needed. We researched the latter type for a cost-effective smart insole using piezo-resistive elastic polymer. In addition, using the castability of the polymer, the insole is molded to release the foot fatigue through forming a swollen region. The cell-to-cell leakage currents are removed using separated bottom electrodes and the material characteristics are examined. Also, the 16 insole data are interpolated to form a larger continuous and mild pressure distribution image using a direct and fast calculating equation in the visualizing process. 

The remainder of the paper is structured as follows. Chapter 2 describes insole structure and fabrication processes. Chapter 3 describes insole material characterization. Chapter 4 describes a data visualization scheme and some results. In chapter 5 we summarize our results.

## 2. Insole Fabrication 

To produce a smart insole that can measure plantar pressure distribution while maintaining mechanical buffering functions, we fabricated two elastomeric polymer layered insole with two electrode layers. Figure 2. shows the insole structure and pressure measuring circuit. The top layer is a non-conductive PDMS (poly-dimethyl-siloxane) layer. The central part of the upper PDMS layer is swollen upward to match the human plantar arch. Below it there is a submerged top electrode. The third layer is CNT (carbon nano-tube) dispersed PDMS layer having piezo-resistivity [8,9,10,11]. The third layer is contacted downward with the bottom 16-electrode layer for generating 16 plantar pressure signals. This lower PDMS layer and the external resistor R between the 16-channel MUX (multiplexer) and ADC (analog to digital converter) comprises the voltage dividing circuit and performs pressure sensing through voltage sensing. The 104 capacitor (0.1 uF) operates as a lowpass filter.

Figure 3 shows the insole fabrication processes. To make the upper convex part, a concave insole mold reflecting human plantar arch is used as in the flipped mode. Blended PDMS and hardener mixture is poured into the mold and hardened. After hardening, a rib-shaped conductive textile is attached on the flat PDMS surface for the top electrode formation and 0.45 wt% CNT-dispersed PDMS is poured on the top electrode to form the lower piezoresistive insole layer. As a result, the top electrode is submerged within the insole. The lower insole material, which has piezo-resistivity, was prepared by dispersing C&T Solution’ CNT powder to the Dow Corning’s PDMS. The CNT and PDMS mixture was blended in the PCM (planetary centrifugal mixer) machine for 2 h at 750 RPM and another 10 min with hardener added. After pouring the lower PDMS layer, air bubbles are removed for 3 h in a vacuum desiccator. If the entire insole is molded in a single process using CNT-dispersed PDMS, the top electrode formation and the removal of air bubbles contained especially in the upper convex PDMS region would be very difficult. The lower PDMS layer was hardened for 1 h at 70°C. During this second hardening, the two PDMS layers merged into a single substance with the top electrode embedded inside it. The bottom 16 electrodes are made on a separate flexible substrate and stacked up on the hardened insole. Figure 4 shows the processing equipment and sequential insole shape.

Figure 5 shows 16 bottom electrodes and the HC4067 16-channel multiplexing module. The 16 electrode configuration is a 3 × 5 matrix shape with one extra electrode mainly for heel press sensing. The electrodes are made of conductive textile patches and wired using copper tapes. The conductive patches are nearly of equal area and are highlighted by the dashed rectangles. These patches define the insole segmentation and the wiring copper lines are connected to 16 channel multiplexer which is controlled by Arduino. The multiplexer output port is pulled down to GND by a fixed resistor and the voltage divided output is directed to Arduino. 

## 3. Material Characterization

The pressure sensitivity and time response of the CNT-dispersed PDMS piezo-resistive material are characterized here. For this purpose, we fabricated a cylinder of radius 13 mm using the piezo-resistive insole material [12,13]. Figure 6a shows the pressure sensitivity measuring setup. A BLDC (brushless direct current) motor-driven stage moves up and down with the sample loaded on the stage and the sample is compressed and restored between the ceiling and the stage. The sample also constitutes a voltage dividing circuit as the insole sensor in Figure 2. Figure 6b and Figure 5c shows the AD converted voltage and the resistance hysteresis curve with respect to the applied force. The applied force was measured using a TAS606-200 kg load cell and ADC was done with Arduino. The ADC curve shows a high degree of linearity up to 1000 N (1.9 MPa). We believe that this material can be used for measuring the plantar pressure with appropriate signal processing. 

We also checked the response time of the cylindrical sample. Since we have no ideal step force exerting machine within millisecond order we hammered the sample with an electrical switch stacked up on the sample as in Figure 7. The switch is normally open, and is closed when it is hit. When the hammer hits, the electrical switch and the sample both suffer an impulse-like force of around 80 N measured with a very fast load cell not shown in the figure.

Figure 8a shows the time response with hammering. For time characteristic measurement we removed the lowpass filter from the voltage dividing circuit so there seems lots of 60 Hz background noise in the sensor output. Switch and sensor output pulses due to the hammering are shown at the center of the window. The switch waveform is offset by 500 from zero for discriminating from the sensor curve and its amplitude is near 1023 (=5 V) level. The sensor output is not offset from zero and the amplitude is around 250 for the 80 N hammer hit. Figure 8b,c show the time detailed waveforms. The switch pulse shows 3ms rising time and 5ms falling time and the sensor output shows 4 ms rising time(Tr) and 8ms falling time (Tf) approximately. Since the switch waveform represent the applied force to the sample the response time is excellent for human plantar pressure sensing.

## 4. Pressure Visualization

The insole outputs 16 voltage signals for one pressure distribution image. Figure 9 shows the smart insole measurement configuration. The smart insole is placed on the bottom electrodes and contact well when downward pressure is exerted from the upside. The upper common electrode is set to Vcc level to generate voltage divided outputs between the piezo-resistive insole layer and the pulldown resistor located after the multiplexer. The output voltages from the bottom electrodes are sequentially fed to Arduino via the MUX and they are AD converted. The AD-converted data are sent to PC through serial communication and processed for pressure image display.

The received data should be calibrated before data processing for two reasons. First the CNT-dispersed PDMS has point-to-point resistance fluctuation over 10%. Since the CNT is a very long chain particle of wide surface they easily attract each other through van der Waals force [14] so the CNT-dispersed PDMS becomes abruptly viscous when the CNT concentration exceeds the percolation threshold which results in difficulty in insole molding process. In addition, higher CNT concentration leads to smaller fractional piezo-resistivity. So as less CNT concentration as possible is recommended. This dilute CNT concentration slightly above threshold level is the main reason for the resistance fluctuation. Second, the insole top surface is bumpy to accommodate the plantar arch. The plantar pressure spreads irregularly according to the pressed position. So we checked the pressure response of each insole cell defined by the bottom electrode as in Figure 10a. We pressed each cell through an elastic bar of radius 13 mm up to ~120 N (~230 kPa) and measured the voltage ADC values. Figure 10b is the ADC hysteresis curve of each cell. The curves show some fluctuations but the overall characteristics are similar with tolerable linearity. So we tentatively use the ADC values as the pressure values with one-point calibration; that is, the ADC values are normalized by the ADC value at 30N for each cell. Of course, a converting factor should be used for the absolute pressure value.

Based on the 16 calibrated data in one frame we extended the frame size virtually and smoothed the pressure image using a pseudo-interpolation scheme for fast calculation. Figure 11. shows a 3-fold frame extension case for example. First, the calibrated 16 data (left) are spatially mapped to the extended frame (right) as uniformly as possible. In the case of m-fold extension, the mapped pixel indices transform as:(1)im=io×m+rounddown(m2)
where ‘*m*’ means ‘*mapped*’ and ‘*o*’ means ‘*original*’.

Second, the mapped cells operate as source pixels in the extended frame as electromagnetic point oscillators act as wave sources to the neighborhood [15]. The mapped data values are used in calculating other non-source pixel values with two assumptions: (1)The value of a non-source pixel is simple addition of all source pixel values weighted by a proper distance function between the source and the non-source pixel.(2)The influence of source pixel on non-source pixel decreases with increasing distance between the two pixels.

Under these assumptions, we suggest each non-source pixel value in the extended frame as below:(2)$$U_{ns}\left(i,\;j\right)=N\times \left[\overset{s=15}{\underset{s=0}{\sum }}\;U_{s}\times e^{-k*d_{ijs}}\right]$$
where $U_{ns}$(*i*, *j*) is the (*i*, *j*)-th non-source pixel value in the extended frame, *N* is a normalizing factor, *s* is the 16 source pixel indices running from 0 to 15, $U_{s}$ is the *s*-th calibrated source pixel value explained above, *k* is an adjustment parameter, and $d_{ijs}$ is the distance between (*i*, *j*)-th non-source pixel and *s*-th source pixel. According to the above equation, every source value is added constructively, weighted by the exponentially decreasing factors of the distances. This is a very simple and easy equation for tuning the imaging adequacy relative to the usual bilinear or cubic convolution interpolation scheme [16]. 

We checked the impact of *k* value. Since *k* value appears as an exponent in Equation (2) small variation of the *k* value has a profound effect in the displayed image. Figure 12a. shows the test setup which is similar to Figure 10. The pressing bar can press 1 ~ 3 bottom electrodes from minimum to maximum. Figure 12b shows 20-fold extended pressure images with different *k’s* for ~4.6 kgf force exerted on the left upper part of the insole. These figures show critical image spreading effect of *k* value. So *k* value must be chosen carefully for the realistic image display. From these results we can see our visualization scheme works reasonably well when the parameter *k* is chosen appropriately.

Figure 13 shows real human plantar measurement photo and several displays of different foot pressure state with *k* value of 0.18. From this calibrated plantar pressure image and its digital values, our smart insole scheme is expected to be used for the human gait analysis if the wiring lines are replaced by a wireless communication device [17].

We summarize our data capture and visualization scheme as in Figure 14.

## 5. Conclusions

In this paper we fabricated an electrically segmented pressure sensing insole instead of the existing smart insole such as an FSR-assisted structure. CNT-dispersed PDMS, the base material of our insole, is a piezo-resistive elastomeric polymer which can offer not only pressure sensing but also mechanical buffering as the traditional insole. The base material in the cylindrical shape showed force to output voltage linearity up to 1.9 MPa and response time of less than 10 ms. We sectored the insole into 16 cells by using 16 bottom electrodes and each cell constitutes a voltage dividing circuit with a serially connected fixed resistor. The voltage outputs are multiplexed to be sent serially to the Arduino chip and are AD converted. Since the sectored 16 cells have different pressure-to-voltage converting efficiency, we calibrated the 16 data using pre-measured voltage ADC values at some applied force. For the pressure visualization, the calibrated data are used as source pixels in the extended image frame for continuous and mild images. The non-source pixel values are calculated using a simple equation using the source pixel values. The calculated values are exponentially dependent on the cell-to-cell distance so the distance coefficient parameter *k* should be chosen carefully. On the other hand, we can easily adjust the image appearance by adjusting the *k* value. Our inexpensive smart insole is expected to be used in the human gait measurement with the installation of a wireless communication device.

## Figures and Tables

**Figure 1 sensors-19-00044-f001:**
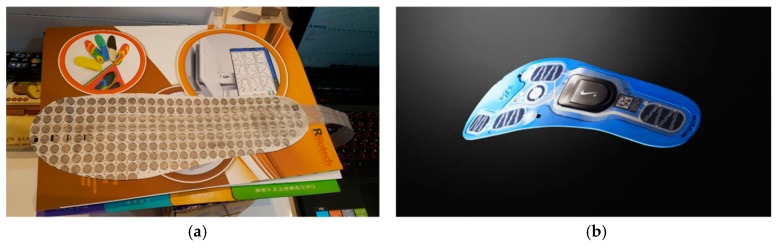
(**a**) FlexinFit pressure sensor, (**b**) Samsung’s smart insole.

**Figure 2 sensors-19-00044-f002:**
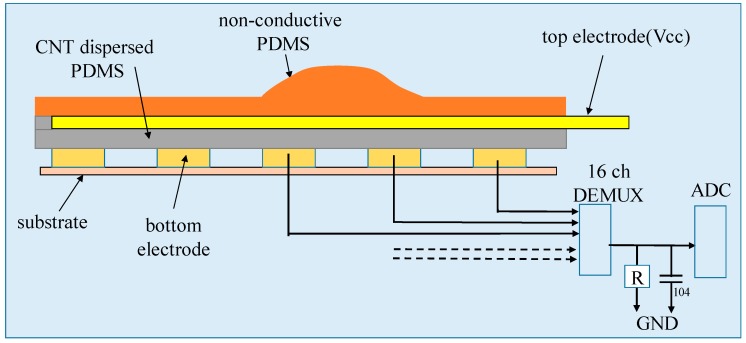
Structure of the polymer-based smart insole.

**Figure 3 sensors-19-00044-f003:**
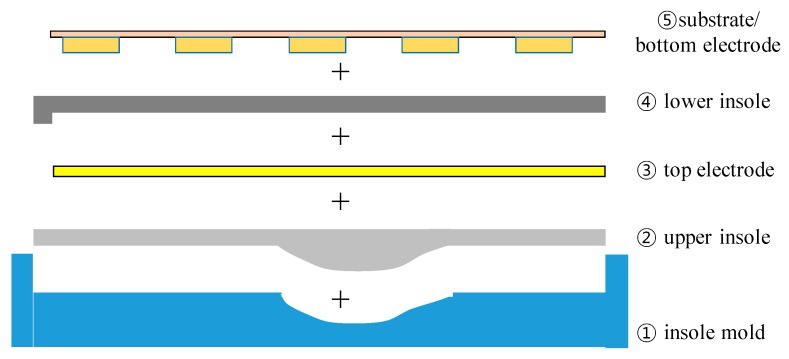
Insole fabrication sequence.

**Figure 4 sensors-19-00044-f004:**
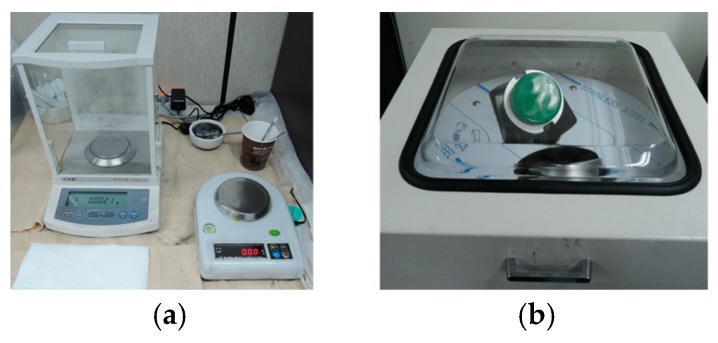
Processing tools and sequential insole shape: (**a**) weighing scale, (**b**) planetary centrifugal mixer (PCM) blending machine, (**c**) insole mold, (**d**) upper poly-dimethyl-siloxane (PDMS) layer with top electrode, (**e**) final insole top view, (**f**) final insole side view.

**Figure 5 sensors-19-00044-f005:**
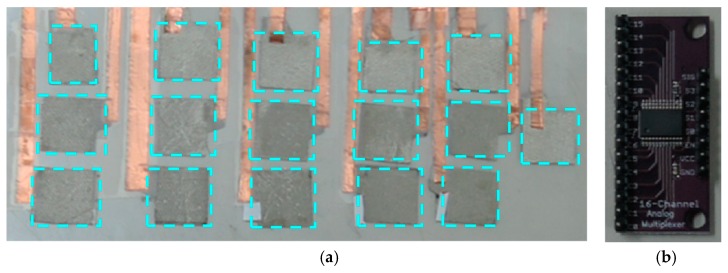
Bottom electrodes and multiplexer: (**a**) 3 × 5 + 1 bottom electrode array, dashed rectangle highlight the electrode patches (**b**) 16-channel multiplexing module.

**Figure 6 sensors-19-00044-f006:**
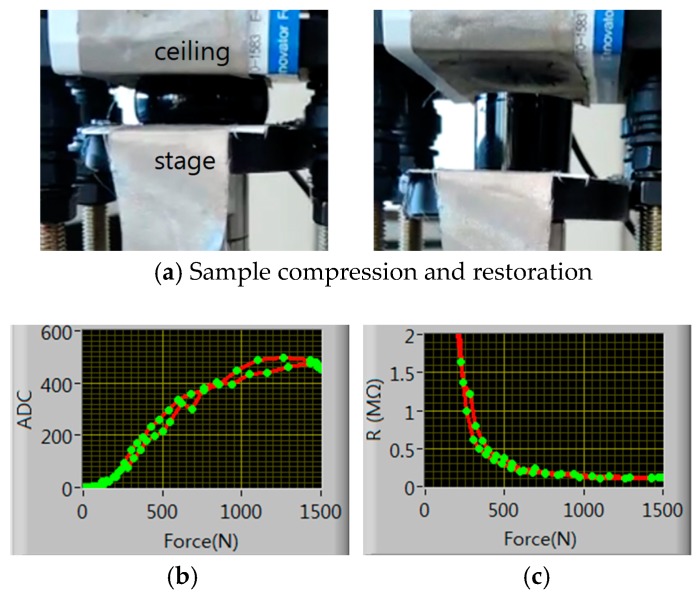
(**a**) Force sensitivity measurement setup, (**b**) force- analog to digital converter (ADC) hysteresis curve, (**c**) force-resistance hysteresis curve(c).

**Figure 7 sensors-19-00044-f007:**
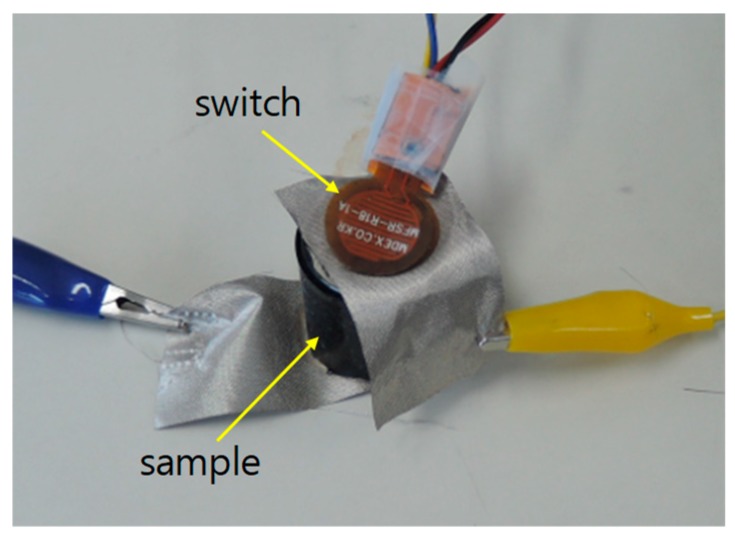
Stack of force indicating switch and cylindrical sample for response time test.

**Figure 8 sensors-19-00044-f008:**
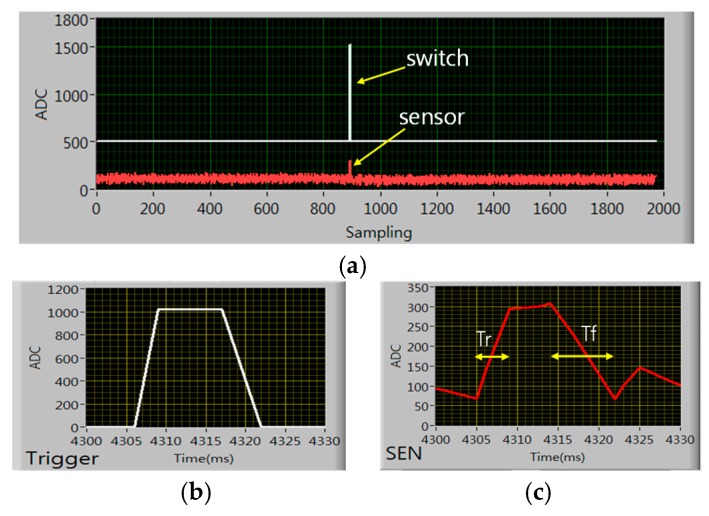
Time response with hammering: (**a**) switch and sensor outputs due to hammering (**b**) switch output, (**c**) sample output waveform.

**Figure 9 sensors-19-00044-f009:**
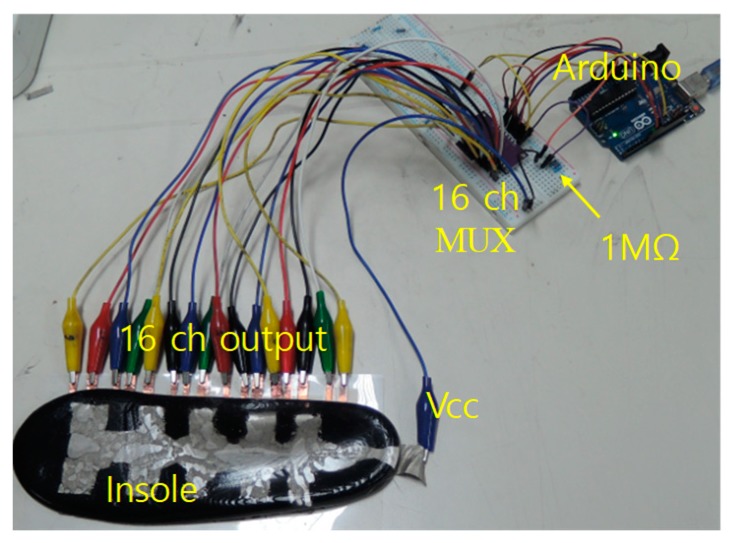
Measurement configuration of the smart insole using Arduino and 16-channel multiplexer.

**Figure 10 sensors-19-00044-f010:**
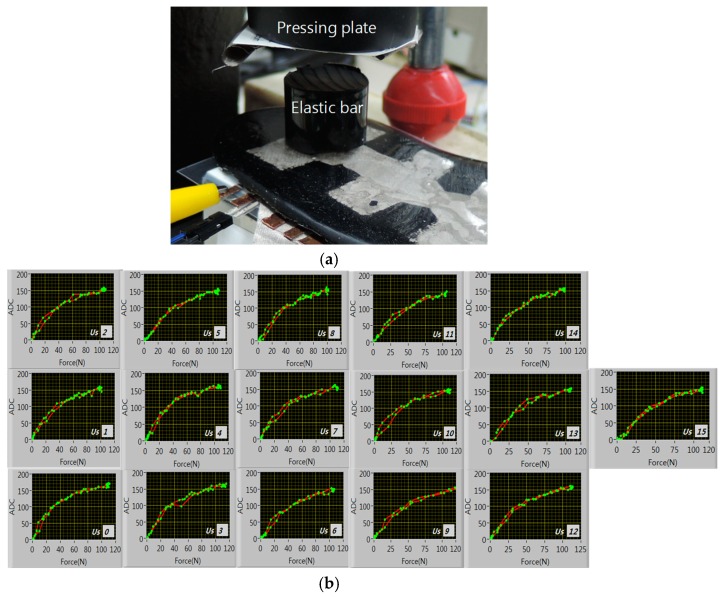
Insole cell calibration setup: (**a**) cell testing setup, (**b**) the ADC hysteresis curves of 16 insole cells.

**Figure 11 sensors-19-00044-f011:**
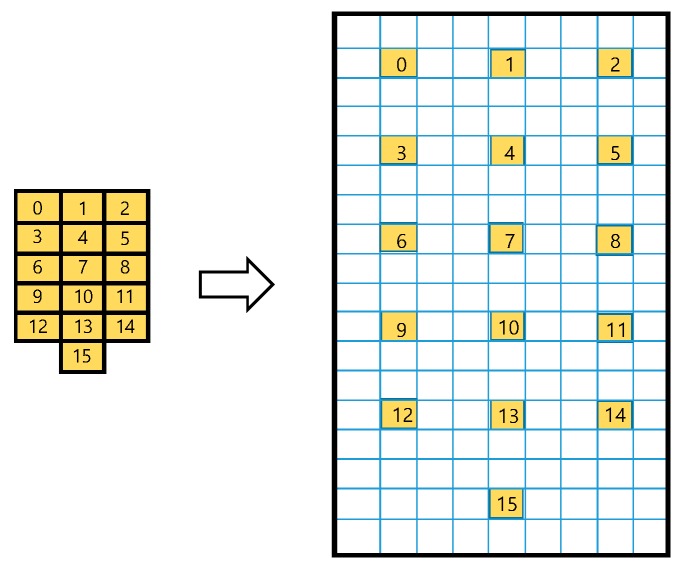
Three-fold frame extension example: Source cells are spatially mapped to the extended frame as uniformly as possible.

**Figure 12 sensors-19-00044-f012:**
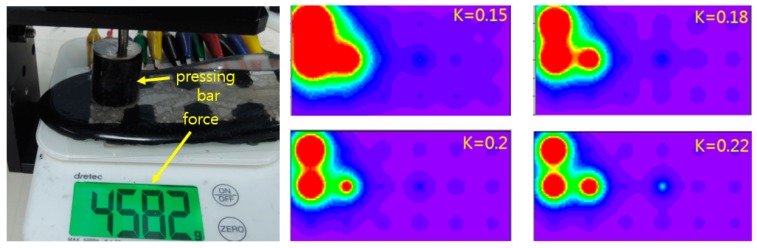
*k’s* characteristic test setup and pressure images with *k* values of 0.15, 0.18, 0.2 and 0.22 for 4.5 kgf force exerted on the left-upper part of the insole.

**Figure 13 sensors-19-00044-f013:**
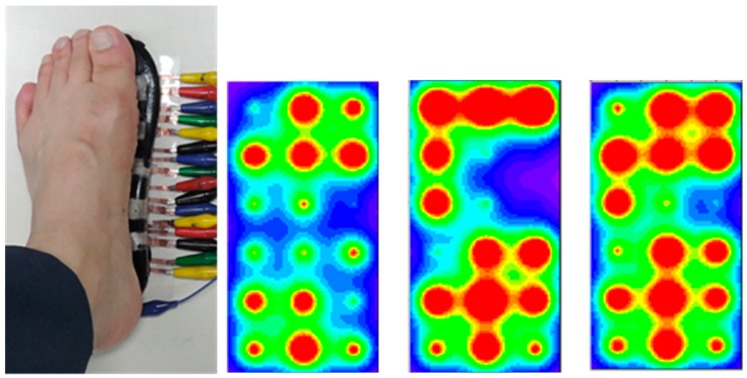
Human plantar measurement photo and the calibrated images of different foot pressure states with *k* value of 0.18.

**Figure 14 sensors-19-00044-f014:**
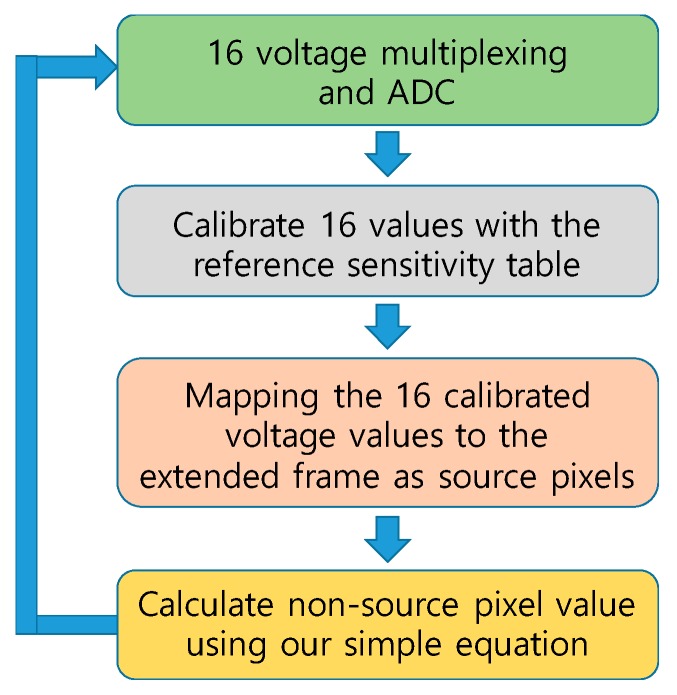
Insole data acquisition and visualization scheme.

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
