# Peer review of "Characterization of Elastic Polymer-Based Smart Insole and a Simple Foot Plantar Pressure Visualization Method Using 16 Electrodes"

_sensors, 2018, doi:10.3390/s19010044_

Reviewer 1 Report

In my opinion, this study is overall interesting and has a good application potential.

However, the methodological approach presents some weaknesses:

57 - the characterization of the piezoresistive effect with respect to the percentage of dispersed CNT should be investigated before anything else and should be shown in the paper (maybe also with respect to other environmental factors)

109-110 What does "good linearity" mean? It would be good to show the graph and indicate the regression function.

122-126 The approach is very practical but not very scientific. The tests should be conducted with appropriate instruments.

In the absence of a more scientific approach, I suggest to modifying the title by highlighting the preliminary character of this work and referring to the future a methodologically more correct approach.

Author Response

57-: Although we measured the piezoresistivity only at one concentration of 0.45wt% (Fig. 5b) it’s well known fact that the fractional piezoresistivity of CNT dispersed polymer varies reversely depending on the CNT concentration. Fig. 2a of Ref. 9(M. Amjadi et al.) shows one example.

109-110: To show the linearity clearly we redraw the Fig. 5b as force-ADC hysteresis curve and reduced the claimed linear range from 1500N to 1000N.

122-126: For accurate time response measuring such as micro-second or nano-second range, ideal step force should be exerted. But we are interested in the ~10ms range, so we think that the exerted force shown in Fig. 7b may be safe for this purpose. 

The following title seems more suitable such as “Characteristics of elastic polymer based smart insole and simple visualization method using 16 electrodes” and we changed.

Reviewer 2 Report

The manuscript provides technical information to develop smart-insole for foot pressure measurement. If insole can maintain the rigid structure to assist walking while continuously recording foot pressure, it will be indeed an innovative and practical technology to utilise foot pressure information in our daily living. The manuscript, however, follows an unorthodox style with a number of editorial issues to be corrected. It is also not experimental with no statistical comparison or testing; on the other hand, very detailed technical information about ‘how to make smart-insole’ has been provided. Following comments are believed to help improve the current manuscript.

General

Manuscript is extremely technical from the beginning to the end, which may not be easy to follow for many readers except a few who are working on this exact topic. Introduction should be re-written to clarify the overall topic in a more general way before getting too much into depth. It is interesting to see all the technical information but some of them seem to be not based on previous work. Justification is required for some details such as the use of filtering or the areas of foot pressure mapping.

Specific

n  Please spell out all abbreviations when used for the first time. (e.g. PDMS, CNT, FST etc)

n  Please use more punctuation to make it readable. (e.g. line 27-29: Among these since human gait pattern may reveal potential sickness and intrinsic healthcare related habits expensive and burdened gait pattern measurement and analyses have been permed in the professional Gait Laboratory).

n  Please correct grammatical errors. (e.g. line 45: ~as following.; line 46: ~fabrication process, section 2.2… and many more).

n  Line 54: “The upper surface of top layer is curved upward to accommodate human plantar pressure.” It is hard to understand what this means. Addition of this arch-support? may alter foot pressure distribution.

n  Line 91-92: Is this 16-electrode configuration orthodox? Does this follow any previous work? Please justify how these 16 areas are divided. Are they all equal in size?

n  Line 125-126: “When the hammer hits, the electrical switch and the sample both suffer impulse like force of around 80N.” What does this mean? Why is it 80N? Doesn’t it depend on how hard it is hit?

Author Response

Abbreviations are spelled out at the first appearance in the text.

Punctuations are added at some lines.

Grammatical errors are checked once again.

Line 54: “curved upward” means “swollen”. This structure may alter foot pressure distribution so we added each cell’s responsivity which are segmented by the 16 electrode(see Fig. 9).

Line 91-92: This is not an orthodox way but a tentative approach. Fig. 1 and Fig. 4 reveals the segmentation algorithm. The segmented areas are defined by the bottom electrodes and they are nearly equal. 

Line 125-126: The whole setup of Fig. 6 is placed on a fast force measuring equipment based on a commercial FSR which is not shown in the figure. We recorded the exerting force during the hammer hitting experiment and the maximum recorded force was 80 Newton.

Global : - Following an orthodox way we changed the materization curve to force-ADC  

                hysteresis curves

             - We revised the introduction to claify the paper position.

             -  Since our paper is preliminary research using elastic piezoresistive polymer,

                statistical approach is deficient. So we changed the paper title to       

        " Characterization of elastic polymer based smart insole and a simple foot plantar pressure visualization method using 16 electrodes".

Reviewer 3 Report

This paper present the development of a pressure-sensing insole. However, lack of sufficient background/references in the introduction and lack of comparing study insole results to existing insole products and insoles presented in other papers does not allow for an evaluation of the potential value of the developed insole. 

MAJOR ISSUES:

1) While some relevant background information is presented in the introduction, the authors do not provide a clear and complete summary of existing insole products (marketed and currently researched). This leaves the reader with insufficient context to properly assess the potential viability, strengths, and weaknesses of the study insole.

2) Characteristics of the sensor have been assessed (hysteresis curve, time response, force measurement). However, none of these tests seem to have been done in a consistent or accurate enough way to provide reliable sensor specifications. Furthermore, none of these test results were compared to similar values for existing insoles. Instead statements like "seems relatively suitable for plantar pressure measuring" are used, which are not convincing or useful for a reader. 

3) Cross talk between individual sensors, capturing all applied force when using a non-matrix/grid sensor (like the F-Scan insole), and accuracy of force measures when the insole is used in a shoe (bending, folding, etc.) are issues that need to be studied and addressed when developing an insole. While the cross talk was addressed to some degree, this analysis does not seem to be  complete and ends with "k value must be chosen carefully". Since this is your custom sensor, you should choose the 'best' k value for typical gait (or the most appropriate k level dependent on body weight) and provide accuracy levels for this k value. 

4) Finally, since there are existing, marketed insoles and other researchers are developing insoles similar to this one, it is important that your paper presents the need or advantages of your sensor. 

5) There are some issues with spelling and grammar throughout the paper.  

Author Response

We revised the introduction to claify the position of our paper and changed the paper title to "Characterization of elastic polymer based smart insole and a simple foot plantar pressure visualization method using 16 electrodes".

We revised the material characterization to Force-sensor ADC hysteresis curves in Fig. 5 and Fig. 9.

The crosstalk said in line 40 (cell interference, revision 0) is hardware crosstalk coming from the leakage currents between the segmented insole sub parts as in Ref. 7(A. M. Tan et. Al). Since we use 16 separate bottom electrodes we have no such type of crosstalk. The ‘k’ value is only a display parameter and has no significance on gait analysis. It’s ok if it is selected for mild and realistic pressure image.

Our research is a preliminary result and distinctive in that piezoresistive polymer is used as a pressure detecting insole itself without hardware crosstalk. We think this is the major advantage.

We checked the spelling and grammar once again.

Reviewer 4 Report

Authors presented a smart insole with 16 pressure points based on two elastomeric and two electrode layers. The approach and insole development is interesting, but the authors should improve the presentation quality following the suggestions listed below:

-Define the acronyms PDMS, CNT, FSR (and all other used acronyms) in the abstract and main text.

- Authors should present the calibration curve force x Voltage (or ADC) for all the 16 measurement points, so the readers can verify, at least, the sensitivity and linearity of each sensor.

- This reviewer misses practical applications with the proposed insole. It should be used at least for plantar pressure mapping of different users in order to show the functionalities of the insole. This is a critical point, since the authors presented the development of the insole, but do not explore in detail the applications of the proposed device.

- The authors should check the whole manuscript for grammar errors and typos.

- Authors should carefully check the manuscript organization. Where are sections 3 and 4? They jump from section 2 to section 5, showing a lack of attention on the paper preparation.

- In addition, authors should consider changing the paper organization, by creating a section for the description of the experimental setup and all tests made. Another section with the characterizations of insole material and the 16 sensors (showing the calibration curve of each one, please) and a section with the results for the insole application, similar to the ones shown in Figure 12.

Author Response

Abbreviations are spelled out at the first appearance in the text.

Force-ADC hysteresis curve of the 16 segmented cells are added in Fig. 9.

We think the main application will be gait analysis but this paper is a preliminary research using piezoresistive polymer as insole itself. So applications to other people sre missed at this paper and we changed the paper title to “Characteristics of elastic polymer based smart insole and simple foot plantar pressure visualization method using 16 electrodes”

We checked the spelling and grammar once again.

& 6. Thanks. chapters are examined and rearranged.

Round  2

Reviewer 3 Report

While some improvements have been made to the paper, some of my key concerns have not been sufficiently addressed.

1) A summary of existing insole products (marketed and currently research) is still not provided. This means the reader still does not have sufficient context to properly assess the potential viability, strengths, and weaknesses of the study insole.

2) This leads directly to an inability to properly interpret the results. Comparison to other insoles or force plates or gait characteristics (i.e., specific values/parameters)) are needed to interpret whether measured linearity, response time are appropriate (for example).

3) The lack of comparison to other insoles makes it difficult to interpret the potential advantages/impact/reader interest in this new insole design. 

Author Response

We compiled existing insole products.

Our insole sensor differs from the conventional insole sensors in its concept.

So, it is difficult to compare to each other.

- All the comments of the reviewer are related issues. So we reply with an overall explain as below  with the corresponding revision of the introduction.

- A wide summary of the existing insole is not  provided but two examples are addressed in the revised introduction.

The  conventional smart insoles are mostly FSR integrated  insoles but our approach is an insole using pressure sensing material.  This is very rare case of smart insole said only in ref. 7 up to our  knowledge. So we compared mainly the difference between ref. 7 and this  paper (cell to cell leakage current).

Reviewer 4 Report

The authors addressed all my comments and concerns.

Author Response

Ok Thank you.